# A Deep Insight into the Diversity of Microfungal Communities in Arctic and Antarctic Lakes

**DOI:** 10.3390/jof9111095

**Published:** 2023-11-09

**Authors:** Alessia Marchetta, Maria Papale, Alessandro Ciro Rappazzo, Carmen Rizzo, Antonio Camacho, Carlos Rochera, Maurizio Azzaro, Clara Urzì, Angelina Lo Giudice, Filomena De Leo

**Affiliations:** 1Department of Chemical, Biological, Pharmaceutical and Environmental Sciences, University of Messina, Viale F. Stagno d’Alcontres, 31, 98166 Messina, Italy; 2Institute of Polar Sciences, National Research Council (CNR-ISP), Spianata S. Raineri 86, 98122 Messina, Italyangelina.logiudice@cnr.it (A.L.G.); 3Stazione Zoologica Anton Dohrn, National Institute of Biology, Sicily Marine Centre, Department Ecosustainable Marine Biotechnology, Villa Pace, Contrada Porticatello 29, 98167 Messina, Italy; 4Cavanilles Institute of Biodiversity and Evolutionary Biology, University of Valencia, C/Catédratico José Beltrán, 2, E46980 Paterna, Spain

**Keywords:** Antarctica, Arctic, polar lakes, extreme environments, fungal community, fungal diversity, fungal ecology, next generation sequencing

## Abstract

We assessed fungal diversity in water and sediment samples obtained from five Arctic lakes in Ny-Ålesund (Svalbard Islands, High Arctic) and five Antarctic lakes on Livingston and Deception Islands (South Shetland Islands), using DNA metabarcoding. A total of 1,639,074 fungal DNA reads were detected and assigned to 5980 ASVs amplicon sequence variants (ASVs), with only 102 (1.7%) that were shared between the two Polar regions. For Arctic lakes, unknown fungal taxa dominated the sequence assemblages, suggesting the dominance of possibly undescribed fungi. The phylum *Chytridiomycota* was the most represented in the majority of Arctic and Antarctic samples, followed by *Rozellomycota*, *Ascomycota*, *Basidiomycota*, and the less frequent *Monoblepharomycota*, *Aphelidiomycota*, *Mortierellomycota*, *Mucoromycota,* and *Neocallimastigomycota*. At the genus level, the most abundant genera included psychrotolerant and cosmopolitan cold-adapted fungi including *Alternaria*, *Cladosporium*, *Cadophora*, *Ulvella* (Ascomycota), *Leucosporidium, Vishniacozyma* (*Basidiomycota*), and *Betamyces* (*Chytridiomycota*). The assemblages displayed high diversity and richness. The assigned diversity was composed mainly of taxa recognized as saprophytic fungi, followed by pathogenic and symbiotic fungi.

## 1. Introduction

The Polar regions (both Arctic and Antarctic) share the common characteristics of extremely harsh climate and offer us a unique and irreplaceable platform to discover and study extremophilic organisms [1,2]. A major feature of Arctic landscapes is the large number of lakes and ponds, which in some regions can cover up to 90% of the total surface area [3]. They contribute significantly to Arctic biodiversity, offering a diverse range of habitats for aquatic organisms, from microorganisms to animals and plants, and providing food and freshwater to migratory nesting birds, resident animals, and humans [4]. Also, the Antarctic continent and sub-Antarctic islands represent some of the most diverse and interesting lake districts on the planet [5]. Apart from subglacial lakes, here, lake ecosystems are found in the limited ice-free areas or oases, where microbial communities of viruses, archaea, bacteria, microalgae, and fungi represent the largest reservoir of biodiversity [6].

Polar regions host an extraordinary diversity of lake types, ranging from freshwater to hypersaline, from highly acidic to alkaline and from perennially ice-covered waters to concentrated brines that never freeze. Many polar lakes are ultra-oligotrophic or extremely unproductive but lakes highly enriched by animal or human activities can also be found [7].

High latitude lakes have broad global significance, acting as residences of unique species and communities and as early detectors of environmental change. Multiple stressors associated with local and global human impact such as contaminant influxes, increased exposures to ultraviolet radiation and climate change have a striking impact on these aquatic ecosystems. In fact, even small changes in physical, chemical or biological characteristics can be amplified into major shifts in limnological properties and in lake ecosystem structure and functioning [8,9,10,11]. The combination of the above-mentioned features makes polar lakes interesting and unique environments to study the taxonomy and ecology of microbial communities living under extreme conditions.

Among the different microbial groups present in polar lakes, fungi are the biggest osmotrophic specialists, producing a plethora of secretory enzymes and obtaining nutrients through extracellular digestion and endocytosis. Thanks to diverse metabolic strategies and high morphological diversity, fungi have conquered numerous ecological niches and have shared various interactions with other living organisms and inorganic surfaces. In fact, fungi are found virtually in all environments throughout the globe, including extreme environments, such as torrid and polar deserts [12,13], hypersaline salterns [14], and deep-sea [15].

The global diversity of fungi was first estimated by Hawksworth [16] to be 1.5 million species. However, the increasing development of DNA sequencing technologies which occurred in the last ten years led to expanding the estimates of fungal species numbers to 2.2–13.2 millions. Despite this high estimate diversity, to date, only around 150,000 fungal species have been described [17]. Fungi constitute a well-founded component of terrestrial ecology due to more than 100 years of research that has highlighted their role in biogeochemical cycling and promoting biodiversity [18], while aquatic ecosystems, in contrast, were long overlooked as fungal habitats. However, fungal diversity, quantitative abundance, ecological functions and, in particular, their interactions with other microorganisms remain mainly speculative, unexplored and missing from current general concepts in aquatic ecology and biogeochemistry [19].

Most previous studies of polar lake fungal communities have used traditional culture-dependent methods [1,10,20,21,22,23,24] which do not reveal the full complexity of the resident fungal diversity. Recent applications of metabarcoding approaches have focused especially on sediment of lakes of maritime Antarctica [25,26,27] and there is only one study on water samples from lakes of continental Antarctica [28]. Very few metagenomic studies have been conducted on Arctic lakes, exclusively from water samples [2,29,30].

In this paper, we used a metabarcoding approach to study the microfungal diversity both in water and sediment sampled from Arctic and Antarctic lakes, to study the fungal community composition of these two different lake matrices in both Polar regions deeper, and to better understand their ecological roles in such extreme environments.

## 2. Materials and Methods

### 2.1. Study Sites and Samples Description

Two sampling campaigns were conducted between 5 and 18 August 2021 in the area of Ny-Ålesund (Svalbard Archipelago, High-Arctic Norway) and between 25 January and 1 February 2022 in Livingston and Deception Islands (South Shetland Islands, Antarctica), respectively. Water and sediment samples were collected from the littoral zone of five Arctic lakes which include Solvannet (L1), Glacier (L2), Knudsenheia (L3), Storvatnet (L4), Tvillingvatnet (L5), and five Antarctic lakes including Argentina (LA), Sofia (LS), Balleneros (LB), Telefon (LT), and Zapatilla (LZ). At each sampling point, the physical–chemical parameters were also measured, particularly temperature, pH, Oxygen (O_2_%), and conductivity (uS/cm), using manual field probes (a CyberScan PC 300 probe was used for Conductivity, pH, and temperature, while oxygen concentration was measured by a Hanna HI9143 probe). Sampling locations and physical-chemical data for each sampling point are shown in Table 1 and Figure 1.

Water samples (n = 10, identified with “w”) were manually collected using a presterilized 2 L-plastic bottle and immediately transported to the labs of the research station to be processed. Here, 1 L of water samples were filtered on polycarbonate membranes (diameter 47 mm; 0.22 µm pore size), in triplicate, and immediately frozen at −20 °C for transport until sample processing in the laboratory at CNR-ISP of Messina, Italy. Surface sediment samples (n = 10, identified with “s”) were collected at the interface water sediments (water depth of 30–60 cm). The first 10 cm of the surface sediment were sampled using a pre-cleaned scoop and pre-sterilized plastic containers, and immediately transported to the labs of the research station where they were directly stored at −20 °C until sample processing. Samples were than processed in the laboratory at CNR-ISPof Messina, Italy.

### 2.2. Total DNA Extraction, Bioinformatic Analyses and Fungal Identification

Total DNA was extracted from membranes and from 1 g of sediment using the DNeasy^®^ PowerSoil^®^ Pro Kit (Qiagen, Hilden, Germany) according to the manufacturer’s instructions. Before the DNA extraction, the samples were gradually thawed at 4 °C. In all DNA extraction steps, we proceeded under strict control conditions within a laminar flow hood to recover the fungal DNA and avoid contaminations. DNA concentrations and purity were quantified by a NanoDrop ND-1000 UV–Vis spectrophotometer. Extracted DNA was used as a template for generating PCR amplicons. The internal transcribed spacer 2 region (ITS2) of the nuclear ribosomal DNA was used as a DNA barcode for molecular species identification. Fungal ITS2 was amplified using the following primers: IlluAdp_ITS31_NeXTf 5′-CATCGATGAAGAACGCAG-3′ and IlluAdp_ITS4_NeXTr5′-TCCTSCGCTTAT TGATATGC-3′ [31]. Sequencing was performed using the Illumina MiSeq platforms, in paired-end form, following the standard protocols of the company EurofinsEurope Services (Germany). FastQC was used to check the quality of raw sequences [32]. Sequences were preprocessed, quality filtered, trimmed, de-noised, merged, modeled, and analyzed by R package DADA2 [33] to infer amplicon sequence variants (ASVs), i.e., biologically relevant variants rather than an arbitrarily clustered group of similar sequences. Particularly we filtered reads by length (minimum length between 150 and 140 bp), by ambiguous bases (no reads with N base were maintained in the analysis), and all sequences were trimmed at the ends after quality control (trimLeft = 17, trimRight = 15). During the analysis, filters for reducing replicate, length, and chimera errors were also applied. Fungal taxonomy annotation was performed using the ITS fungal database, UNITE—Unified system for the DNA based fungal species linked to the classification [34], formatted for DADA2, offering an updated framework for annotating fungal taxonomy (unified system for the DNA based fungal species linked to the classification, identity, similarity used cutoff 95%). Finally, a manual inspection was done, and sequences with an abundance of below 0.1% were considered together in the minor groups of retrieved fungi. All sequences have been submitted to the National Center for Biotechnology Information (NCBI) under the BioProject PRJNA1000778.

### 2.3. Fungal Diversity, Distribution and Predictive Functional Profiling

The numbers of reads obtained for each water and sediment sample were used to quantify taxon alpha diversity, richness and dominance, using the following indices: Fisher, Shannon, Chao1, ACE, Simpson and InvSimpson. Venn diagrams were prepared using the retrieved ASVs using InteractiVenn online tool [35] to compare the fungal assemblages present in the lake samples. Functional assignments of fungal ASVs at species and genus levels were analysed using tool FUNGuild [36]. FUNGuild v1.1 is a flat database hosted by GitHub (https://github.com/UMNFuN/FUNGuild (accessed on 20 May 2023)), accessible for use and annotation by any interested party under GNU General Public License.

### 2.4. Statistical Analyses

To compare the fungal community compositions across groups of samples, the Bray–Curtis similarity analysis was performed and similarity matrices were used to obtain dendrograms using R base packages. Principal component analyses (PCAs) were performed using the factoextra R package, on data from selected physical and chemical properties of sediments and waters, and the relative abundance of significant fungal groups. The environmental variables used in these analyses were as follows: oxygen (O_2_%), temperature (°C), water electrical conductivity (Cond uS/cm), and pH.

## 3. Results

### 3.1. Influence of Environmental Paramenters in Lake Clustering

In the principal component analysis of the physico-chemical parameters, it was possible to observe that lakes were completely separated by environmental factors. In fact, the Arctic and Antarctic samples were completely distinct and generated two different groupings, with the only exception represented by L3, which was strictly related to the conductivity, due to its closeness to the sea cost and therefore it is considerably influenced by salt water. Results are shown in Figure 2.

### 3.2. Fungal Taxonomy

A total of 1,639,074 DNA merged reads of good quality were detected in the water and sediment samples from the ten lakes, with an average length of between 350 and 450 bp, representing 5980 ASVs. Unfortunately, two of the sequenced samples (i.e., L1-s and L2-s) did not produce good results in the first enrichment steps, with creation of a low-quality library and subsequently the impossibility to continue with the NGS sequencing. This was probably due to low the ITS DNA quantity, even though the total extracted DNA showed high concentration and good quality (Appendix A). In the Arctic lakes, the analysis of phyla showed that almost in all samples an average of 50% ASVs were related to Fungi_phy_Incertae_sedis, consisting of ASVs whose taxonomical relationships and positions are unknown or not defined. The dominant phylum was represented by *Chytridiomycota* in all samples examined, except for L4-s, where the phylum *Rozellomycota* was most represented. The highest value of ASVs assigned to *Chytridiomycota* was observed in L4-w (46.2%) and the lowest value in L5-w (4.9%). *Rozellomycota* was the second most represented phylum, but with an uneven distribution: the highest abundance value was 20.5% (L4-s) and the lowest was 0.16% (L1-w). The phyla *Ascomycota* and *Monoblepharomycota* were retrieved with percentages higher than 1% only in L5-s (5.49%) and L3-s (2.41%), respectively. Finally, the phylum *Basidiomycota* was found with percentages higher than 1% in L2-w (1.52%) and L5-s (3.76%). The fungal community structure in water and sediment samples of the Arctic lakes is shown in Figure 3.

Overall, the same phyla were retrieved in Antarctic samples, but with some crucial differences. The results are shown in Figure 4. First, Fungi_phy_Incertae_sedis showed an average percentage of around 26% and were about completely absent (0.04%) in LA-w, underlining a fungal community composition related mostly to known phyla. In addition, in Antarctic samples *Chytridiomycota* were the most abundant phylum and ubiquitously distributed in all analyzed samples. Their highest value (68.5%) was retrieved of LS-w and the lowest value was obtained for LT-w (12.85%). The phylum *Ascomycota* in Antarctic lakes was retrieved with a higher value than the Arctic samples. It was found in all lakes with a value comprised between 65.8% (water of LA) and 0.45% (LZ-w). Phyla *Rozellomycota* and *Monoblepharomycota* were retrieved with percentages higher than 1% only in LA-s (40.3%) and LZ-w (3.16%), respectively. Finally, the phylum *Basidiomycota* was retrieved in all Antarctic samples with an average percentage of 4.5%. Their highest value was found in LB-w (23.9%) and the lowest in LZ-w (0.06%).

Based on all the retrieved phyla, a principal component analysis (PCA) was performed, and the results were used for selecting fungal taxa that had high variance (*Ascomycota*, *Basidiomycota*, *Chytridiomycota*, *Rozellomycota* and Fungi_phy_Incertae_sedis). With that, then was performed the construction of the PCA to identify groups of samples with similar community compositions (Figure 5).

The PCA showed a distinction between the Arctic and Antarctic samples, in fact six of the eight Arctic samples clustered together and their separation was driven by Fungi_phy_Incertae_sedis. Two exceptions in the Arctic cluster were water from both L4 and L3 which were found to be related to most Antarctic samples and related to the presence of *Rozellomycota*. Finally, two water samples of Antarctic region (LA and LB) were completely separated by other groups and noticeably related to the phyla *Ascomycota* and *Basidiomycota*.

The most abundant genera (percentage above 1% for at least one sample) were summarized in a heat map (Table 2). Overall, a total of 17 and 50 genera were detected in lakes from the Arctic and Antarctic regions (Table 2), respectively. In the Arctic, L5-s showed most of the genus-wide affiliated sequences. *Betamyces* was the most abundant genus in the Arctic lakes, and it was retrieved in all analysed samples. In particular, the highest abundance was retrieved in L5-s (16.6%). The genera *Pseudeurotium* and *Amylocorticiellum* were retrieved with more than 1% of abundance (2.1% and 2.6%, respectively) in L5-s, together with the genus *Leucosporidium* (1.4%) in L2-w. The genus *Pseudeurotium* was present in almost all samples, the genus *Leucosporidium* was present in half of the Arctic samples, instead *Amylocorticiellum* and was only found in L5-s. For Antarctic samples, LB-w showed most of the genus-wide affiliated sequences. *Cladosporium* represented the most abundant genus retrieved, with the highest value that was observed in LA-w (39.5%). Similarly, the genus *Cadophora* was mainly retrieved in LA-w with a percentage of 19.5%, followed by the genera *Malassezia* (8.38%) and *Alternaria* (5.9%), with the latter that was found exclusively in this sample. *Ulvella* was the most abundant genus in LB-w and LB-s (14.2% and 13.3%, respectively), followed by *Vishniacozyma*, which was retrieved with a percentage of 7.5% in LB-w. The genus *Betamyces* was found in all studied samples from Antarctic lakes, exception for LT and from LB-s, with the highest abundance in LZ-s (4.32%). In LS, *Coleophoma* was the most represented genus, both in sediment (6.51%) and water (1.05%). The ascomycetous genus *Metschnikowia* was the most abundant (3%) in LT-w. Interesting to note is that even though Arctic samples showed very few numbers of genera, *Amylocorticiellum, Helicodendron*, *Iodophanus, Scolecolachnum* (all found in L5-s), *Haptocillium* (L4-s), *Knufia* (L3-w), and *Zygophlyctis* (L1-w) were exclusively present in the Artic and completely absent in Antarctica.

### 3.3. Fungal Diversity and Distribution

A total of 3334 ASVs were obtained from the Arctic lake samples (both water and sediment) and, of them, only 3 ASVs (0.09%) were shared between the two matrices (Appendix A). Overall, lakes L4 vs. L5 and L1 vs. L3 shared the highest percentages of ASVs (7.1%and 6.9%, respectively), while L1 vs. L5 shared the lowest percentages of ASVs (2.8%). Similarly, the difference in ASV distribution between water and sediment samples was evaluated, trying to understand if lakes showed similar populations, or water and sediment have different inhabitants. Considering the two different matrices, 1732 ASVs and 1881 ASVs were obtained from Arctic water and sediment samples, respectively, with 279 ASVs (8.36%) being in common. Comparing the ASVs obtained exclusively from water samples from each Arctic lake, L1-w vs. L3-w shared the highest percentage of ASVs, followed by L2-w vs. L5-w and L4-w vs. L5-w (16.5%, 9.3% and 8.2%, respectively), while the lowest percentages of shared ASVs was observed for samples L3-w vs. L4-w (3.2%) (Appendix A). Lower percentages of shared ASVs were obtained from sediment samples L3-s vs. L4-s, L3-s vs. L5-s, and L4-s vs. L5-s (1.6%, 3.4% and 3.8%, respectively) (Appendix A). Antarctic lake samples (both water and sediment) gave a total of 2748 ASVs and, of them, 4 ASVs (0.15%) were shared among samples (Appendix A). The highest percentages of ASVs were shared between LA vs. LS and LB vs. LT (11.6% and 10.2%, respectively), while LT vs. LZ shared the lowest percentages of ASVs (1.9%). In particular, 1270 ASVs and 1843 ASVs were obtained from Antarctic water and sediment samples, respectively, and a total of 365 ASVs (13.28%) were shared between the two matrices (Appendix A). With regard to ASVs obtained exclusively from water samples from each Antarctic lake, LB-w vs. LT-w shared the highest percentage of ASVs (12.5%), while the lowest percentages of shared ASVs was observed for samples LA-w vs. LT-w (1.3%) (Appendix A). For ASVs obtained exclusively from sediment samples of each Antartic lake, LA-s vs. LS-s shared the highest percentages of ASVs, followed by LB-s vs. LT-s (13.4 and 7.2%, respectively). Only 0.5% of ASVs were shared between LT-s and LZ-s (Appendix A). Considering the two Polar regions, of the total 5980 ASVs retrieved, only 1.7% (102 ASVs) were shared between the Arctic and Antarctic samples (Appendix A). The percentage of common ASVs between water and sediment samples were higher than that underlined between the Arctic and Antarctic samples, showing a value of 10.87% (Appendix A). Finally, following the above results, the ASVs were examined separated by region (Arctic and Antarctic) and by matrix (water and sediment). The results are shown in Figure 6. Arctic vs. Antarctic water and Arctic vs. Antarctic sediment showed really low percentage of common ASVs (1.79% and 1.16%, respectively). Put differently, water vs. sediment from Arctic and water vs. sediment from Antarctic region, shared higher number of ASVs (8.36% and 13.28%, respectively).

The diversity indices were calculated for each water and sediment sample, based on final ASVs obtained after bioinformatics analyses (Table 3). Sediment showed slightly higher diversity values (mean of indices value: Chao1 512.9; ACE 513.3; Shannon 3.9; Simpson 0.9; InvSimpson 18.5; Fisher 72.5) if compared with water (mean of indices value: Chao1 354.1; ACE 354.1; Shannon 3.4; Simpson 0.8; InvSimpson 17.7; Fisher 48.2). The highest value of Shannon index was found in water sample of LS (4.82) (Livingston Island, Antarctica) and a comparable value was found also in the sediment of the same lake (4.49). Instead, the lowest Shannon diversity value of was retrieved in the water sample of L3 (Svalbard Island, High Arctic), and also sediment of the same lake showed a value lower than the average sediment Shannon index value (3.17).

### 3.4. Predicitive Functional Profiling of Fungal Communities

In total, a function was assigned to 103 ASVs (accounted for 1.7% of total ASVs). Genera with confidence level of “possible” were classified as “uncertained” and excluded from the functional analyses in this study. Retrieved function were assigned to saprotroph (57%), pathotroph (30.9%) and symbiotroph (12.1%) (Appendix A). For the Arctic samples, the functionality was assigned to a minor number of ASVs, and pathotroph and saprotroph showed 3.03% and 6.7% in L2 and, 12.7% and 18.8% in L5, respectively. Particularly, the saprotrophic function was underlined for the ASV 24 affiliated to the taxon *Betamyces* (L5). Interestingly, it was also observed the assignment of symbiotroph function to a great number of sequences retrieved in LA and related to the taxon *Cadophora*. In general, a predictive ecological function could be assigned to a very low percentage of ASVs. This was probably due to the fact that our samples come from scarcely studied environments and they have communities composed of a great number of organisms not yet identified.

## 4. Discussion

Microorganisms are the dominant life forms in the Arctic and Antarctic regions. Amongst the groups of microorganisms occurring in these regions, fungi are one of the most abundant and better distributed in the various environments, playing a crucial role in the micro- and macro food webs. Particularly fungi inhabiting polar lakes play a key role in biogeochemical cycles and the mineralization of organic matter, which are essential for the balance of micro- and macronutrients in lake systems. Many fungal species display multiple stress tolerance capabilities, surviving the combination of low temperatures, high salinity, pH variation, seasonally high UV radiation and low nutrient availability experienced in different polar lakes [1,6,9]. However, despite their importance, the availability of studies of fungal diversity in polar lakes has increased only in recent years, but it still remains scant and fragmentary. To date, this work represents the first study of fungal communities in water and sediment of both Arctic and Antarctic lakes.

### 4.1. Fungal Diversity

The total fungal community detected through metabarcoding showed, within the analyzed lakes, comparable values of diversity for both Poles (Table 3), underlining no overall difference between the Arctic and Antarctic lakes, although in general, a slightly greater diversity for sediment than for water was observed. Comparable results were obtained by Perini et al. [30], who calculated fungal diversity in waters of a lake from Ny-Ålesund with a Shannon index H′ = 3.27, corresponding to the mean value obtained in this study (H′ = 3.3), while greater values (comprised between H′ = 3.83 and 5.24) were obtained by Zhang et al. [2]. The diversity data of fungal sequence assemblages detected in the sediment of Antarctic lakes LA, LB, LS, LT, and LZ studied here were greater than those reported in previous culture-based studies [1,10,24], and comparable with those reported in DNA metabarcode study by Ogaki et al. [25], de Souza et al. [26], Rosa et al. [27], and Gonçalves et al. [37] for other Antarctic lakes. However, results obtained by de Souza et al. [26] in sediment of Soto Lake, located in Deception Island (Antarctic Peninsula), which hosts three of the lakes examined in the present study (i.e., Balleneros, Telefon and Zapatilla), showed lower diversity indices (Fischer = 10.27). Although comparable diversity was observed between the studied lakes in the two regions (Arctic and Antarctic), a considerable difference was observed in terms of community composition. In fact, only 102 ASVs (out of 5980 ASVs; 1.7%) were shared between the Arctic and Antarctic samples. This result suggests that fungal distribution varies between the lakes in the two Polar regions and each counterpart hosts specific fungal taxa. Not only geographical distance, but also physical-chemical parameters that separate lakes of the two regions (as it is shown in Figure 2) and different sampling time could contribute to shaping the composition of the fungal community in lakes belonging to the two different regions.

### 4.2. Fungal Phyla

In Arctic lakes, unknown fungi dominated the sequence assemblages, with almost half of the obtained ASVs being assigned to Fungi_phy_Incertae_sedis. This assignation suggests the dominance of possibly undescribed fungi, or that these taxa provide examples of sequences not currently included in publicly accessible databases. The problem arises from the scarcity of metabarcode and metagenome studies of fungal communities in these polar ecosystems, in particular for Arctic lakes. This fact is also corroborated by the results obtained by the PCA of each fungal group for each sample (Figure 5), where almost all fungal assemblages in Arctic water and sediment correlated with the group of Fungi_phy_Incertae_sedis. The most represented identified phyla were, in order, *Chytridiomycota*, *Rozellomycota*, *Ascomycota*, *Basidiomycota*, which were commonly reported by Comeau et al. [29], Zhang et al. [2] and Perini et al. [30] in Arctic lakes. Less frequent, instead, were *Monoblepharomycota*, *Aphelidiomycota*, *Mortierellomycota*, *Neocallimastigomycota* and *Mucoromycota* which were never reported in Arctic lakes, but (excluding the phylum *Neocallimastigomycota*) were previously detected in Antarctic lakes [25,26,27,37].

In Antarctic samples, the percentage of ASVs assigned to unknown fungi was lower (average of 26.70%) than the ASVs obtained from Arctic samples, probably due to the higher number of studies of fungal communities in Antarctic lacustrine systems that increased considerably in recent years. *Chytridiomycota* results as the most represented phylum, followed in the order by *Ascomycota*, *Rozellomycota*, *Basidiomycota*, *Monoblepharomycota*, *Mortierellomycota* and *Aphelidiomycota*. Different studies showed that members of the phyla *Chytridiomycota* and *Cryptomycota* (*Rozellomycota*) dominated the fungal community composition in European freshwater lakes [38,39]. Similar results were also obtained in marine and polar freshwater environments [2,29,40] and the recent use of DNA metabarcoding approaches has revealed the presence of *Chytridiomycota* and *Rozellomycota* assigned sequences also in Antarctica, with reports from soil [41], air [42], mosses [43], permafrost [44], marine sediments [45], snow [46] and, recently, in lake sediments [25,26,27] and lake water [28]. The phyla *Rozellomycota* and *Chytridiomycota* have some physiological advantages for inhabiting aquatic ecosystems, including their mobility and capacity to parasitize numerous phytoplankton species such as diatoms, green algae, dinoflagellates and cyanobacteria [47,48]. Furthermore, the *Chytridiomycota* are implicated in a variety of ecological processes, such as the transfer of organic matter from phytoplankton into zooplankton via saprophytic and parasitic activity [49]. Taxa in this group are thought to mediate the transfer of organic matter from phytoplankton to zooplankton via saprophytic and parasitic activity described as the “mycoloop” [49]. The phylum *Ascomycota* was the most represented in water samples of LAwith a percentage of 65.87%. This dominance could be related to the origin of this lake, which derives from the ice-melting of a close glacier. In a recent DNA metabarcoding study [46], it was reported that *Ascomycota* represents the dominant phylum in Antarctic snow. So, by the water supply from the glacier, this phylum could enrich the fungal composition of the lake. In our study the infrequent phylum *Neocallimastigomycota* was found exclusively in Arctic lakes, in water of L1 and in sediment of L3, L4 and L5. This phylum was never reported in studies of polar lakes. The members of *Neocallimastigomycota* are anaerobic-flagellate fungi residing in the rumen and alimentary tract of larger mammalian and some reptilian, marsupial and avian herbivores, where they play an important role in the degradation of plant material [50]. The detection of this phylum in Arctic lakes analysed in this study could be due to the presence of birds or reindeers which visit these lakes.

### 4.3. Fungal Genera

Generally, it is very difficult to reach the genus level for fungi by metabarcoding analyses, and in our case particularly due to the scarcity of information and deposited sequences of fungi in the environments under consideration. In our study, the most represented genera found in Arctic lakes were *Betamyces* (*Chytridiomycota*), *Pseuderotium* (*Ascomycota*), *Amylocorticiellum* and *Leucosporidium* (*Basidiomycota*), while *Cladosporium*, *Cadophora*, *Ulvella*, *Alternaria*, *Coleophoma*, *Metschnikowia* for *Ascomycota* and *Vishniacozyma*, *Malassezia* for *Basidiomycota* were the most represented genera in Antarctic lakes, some of which have previously been reported from different environment in the Polar regions. The genus *Betamyces* (*Chytridiomycota*) was retrieved in freshwater ecosystem in Ny-Ålesund (Arctic) [2] and in sediment of Antarctic lakes, where *Betamyces* spp. dominated the assemblages [37]. The genus includes only one known species, *Betamyces americae-meridionalis*, which was isolated from pollen baits at the Paraná River (Buenos Aires, Argentina) and in soil in Costa Rica [51]. The genus *Pseuderotium* (*Ascomycota*) was previously reported in Arctic aquatic environments (streams, ponds, melting ice water, and estuaries) in the Ny-Ålesund Region using 454 pyrosequencing [2]. In a metabarcode study of a sediment core of Lake Boeckella (Antarctic peninsula) the genus *Pseuderotium*, with the species *P. hygrophilum*, represented one of the most abundant taxa [27]. The genus was isolated from different Arctic and Antarctic environments and substrata, such as lakes [1,10,30], active layer of the ice-free oases in continental Antarctica [52], sponges [53], and soil [54]. The genus *Leucosporidium* was described in the course of investigation of Antarctic heterobasidiomycetes [55]. It comprises 12 species, mostly phychrotolerant or psychrophilic, which occur in plant materials, soils, and marine environments of high and moderate latitudes [56,57,58]. Among *Basidiomycota*, *Leucosporidium* was the most represented genus in fungal assemblages of Arctic freshwater [29] and its isolation from Antarctic lakes was previously reported [10,20,21,26,59,60]. The basidiomycetous genus *Amylocorticiellus* consists of four species with a widespread distibution. The species *Amylocorticiellus mollis* (known in ‘building mycology’ as *Leucogyrophana mollis*) resulted as the dominant wood-decaying fungus in samples taken from wooden historic constructions in Svalbard [61]. *Cladosporium*, *Cadophora* and *Alternaria* are melanized ascomycetous fungi distributed worldwide and occupy various ecological niches. They are known to be able to resist harsh environmental conditions such as high temperatures, scarcity of water, and high UV radiation [62]. Most species of this genus are plant pathogens or endophytes [63,64], wood destroyers [61], and soil inhabitants [65]. Some species of the genus *Cadophora* and *Cladosporium* are psychrotrophs [66,67] and the presence of these genera are well documented both in Antarctica [6,54] and in the Arctic [2,30,68]. *Malassezia* is a lipophilous basidiomycete yeast genus typically associated with vertebrate animals, but culture-independent studies revealed their presence in diverse acquatic and terrestrial ecosystems. In particular, *Malassezia* is reported among the fungal genera with widest distributions across various polar niches [69], having been reported several times by Arctic and Antarctic culture-independent studies [27,30,70,71,72]. The literature about the genus *Ulvella* is very scarce. There is only one species, *U. chlorospila* synonym *Pyrenula chlorospila* Arnold. The genus *Pyrenula* is a group of crustose lichens typically growing on smooth, shaded bark. It comprises 745 named species with worldwide distribution, most represented in the tropics and Europe [73], but never reported from polar environments. *Vishniacozyma* is a cosmopolitan genus, and it has been reported in cold environments around the world, including subglacial ice samples from Svalbard Islands [30], and soil and wood in Antarctica [59,60,70,74]. The ascomycetous yeast genus *Metschnikowia* was frequently reported from polar habitats such as sea ice, invertebrates, macroalgae, marine sediment, sea water [6,75]. *M. australis*, which is considered an endemic Antarctic species, was isolated from biofilms sampled in Lake Kroner (Deception Island) [24]. The genus *Coleophoma* includes species reported as plant pathogens, saprophytes or endophytes for different plant species [76], and it was reported from sediment of Lake Wanda and from moss samples from King George Island, Antarctica [22,43].

### 4.4. Fungal Ecology

In our study, we deeply investigate fungal community composition additionally including the use of functional prediction. As a result, Arctic and Antarctic fungi displayed different ecological roles as saprophytes, mutualists, symbionts, and/or parasites. Saprophytic fungi dominated the assemblages detected in water and sediment of the Arctic and Antarctic lakes examined, followed by plant and animal pathogens and symbionts. Similar results were reported for sediment of lakes on Vega Island, Elephant Island, Deception Island, Jame Ross Island, and Trinity Peninsula [25,26,27,37], all in maritime Antarctica. The same functional ecological profiles were reported in metabarcoding studies of different Antarctic habitats, such as air and snow [42,77], soil [41], freshwater [78] and rock surfaces [79]. According to Schütte et al. [80], fungi present in polar environments display the capability to degrade organic matter at low temperatures, thereby releasing compounds containing carbon, nitrogen, and other elements to other organisms. The dominance of saprophytic fungi in water and sediment of the examined Arctic and Antarctic lakes might indicate, as suggest by de Souza et al. [26] the presence in these environments of a complex saprophytic fungal community that plays a vital role in the decomposition of organic matter under extreme conditions.

## 5. Conclusions

To date, studies on the fungal communities of lake ecosystems in Arctic and Antarctic regions are decidedly few and the majority of them have been based primarily on traditional culture-dependent approaches, thus underlining a still large gap in the understanding of these sensitive environments. This study is the first to compare fungal communities in water and sediment of Arctic and Antarctic lakes. Metabarcoding analysis revealed complex fungal communities in water and sediment of Polar lakes, which may be considered hotspots of fungal diversity, potentially including new and previously unreported species. The results obtained show clearly distinct communities between the analyzed environments, probably due to the different environmental factors and limnological differences between the analyzed lakes, however, showing some common threads. In particular, the most frequently found phyla are generally ubiquitous, even if for the first time the presence of *Neocallimastigomycota* is reported in Arctic samples. Furthermore, the lake water and sediment fungal assemblages were dominated by saprophytes, which may contribute to the decomposition of organic matter under extreme conditions. However, further investigation is necessary to better understand the ecological role of freshwater fungi in polar lakes, and in particular their roles in nutrient cycling.

## Figures and Tables

**Figure 1 jof-09-01095-f001:**
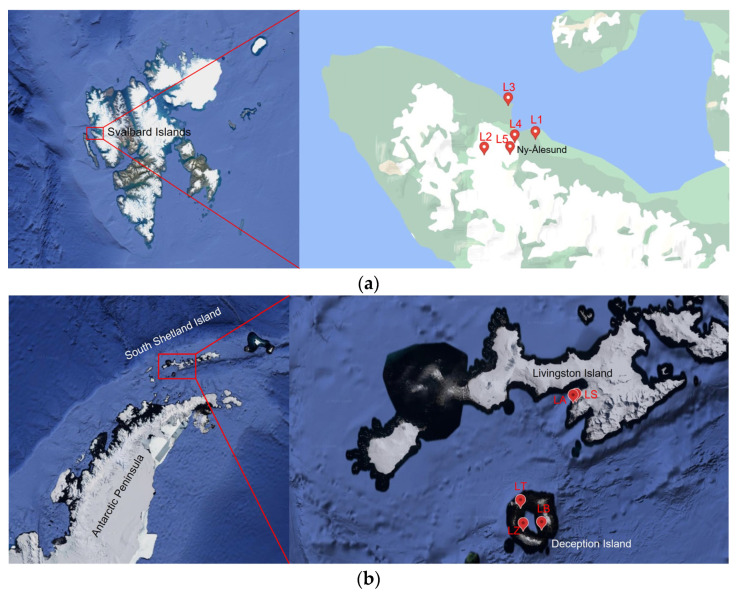
Maps showing the location of the sampling sites in (**a**) Ny-Ålesund area (Svalbard Islands) and in (**b**) Livingston and Deception Islands (Antarctica). L1 = Lake Solvannet, L2 = Lake Glacier, L3 = Lake Knudsenheia, L4 = Lake Storvatnet, L5 = Lake Tvillingvatnet, LA = Lake Argentina, LS = Lake Sofia, LB = Lake Balleneros; LT = Lake Telefon, LZ = Lake Zapatilla.

**Figure 2 jof-09-01095-f002:**
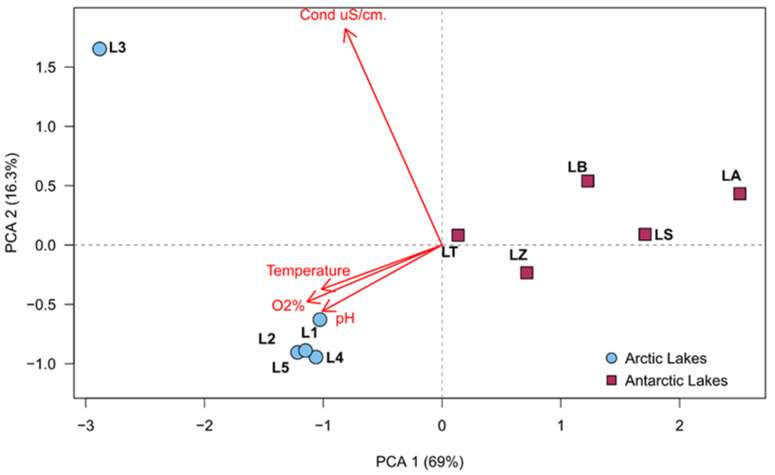
Principal component analysis obtained by recorded environmental parameters, made by factoextra R package.

**Figure 3 jof-09-01095-f003:**
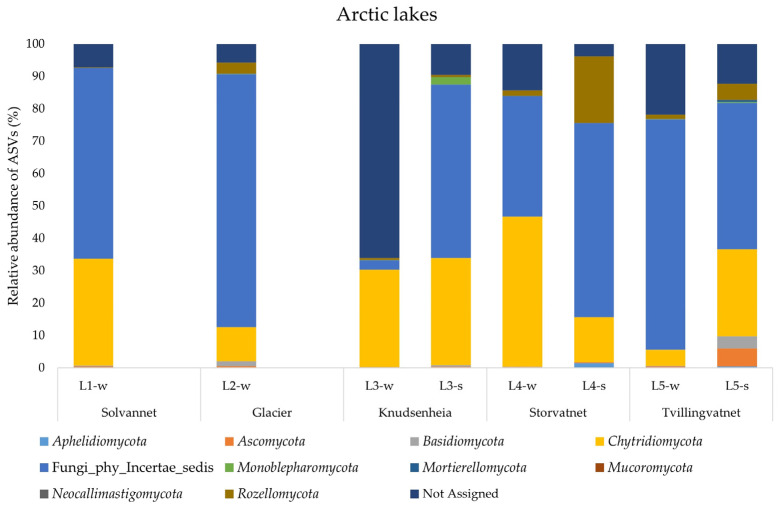
Fungal community structure at the phylum level in the water and sediment samples from Arctic lakes.

**Figure 4 jof-09-01095-f004:**
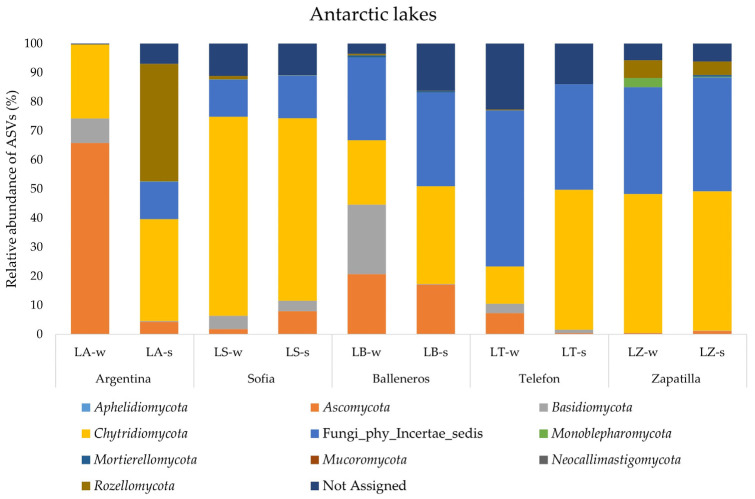
Fungal community structure at the phylum level in the water and sediment samples from Antarctic lakes.

**Figure 5 jof-09-01095-f005:**
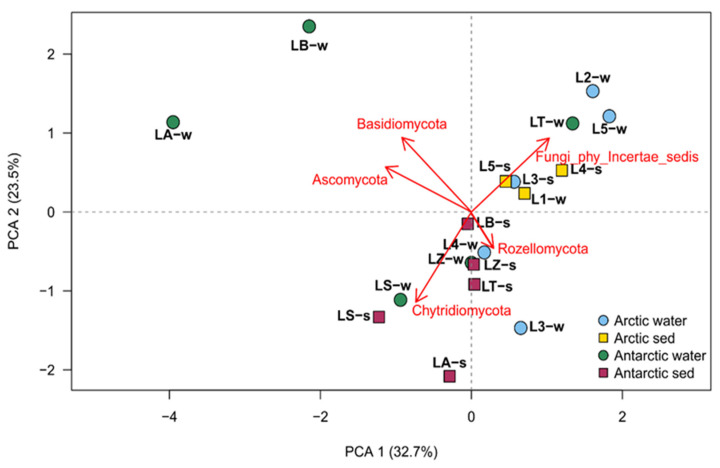
Principal component analysis obtained with the retrieved phyla information parameters, made by factoextra R package.

**Figure 6 jof-09-01095-f006:**
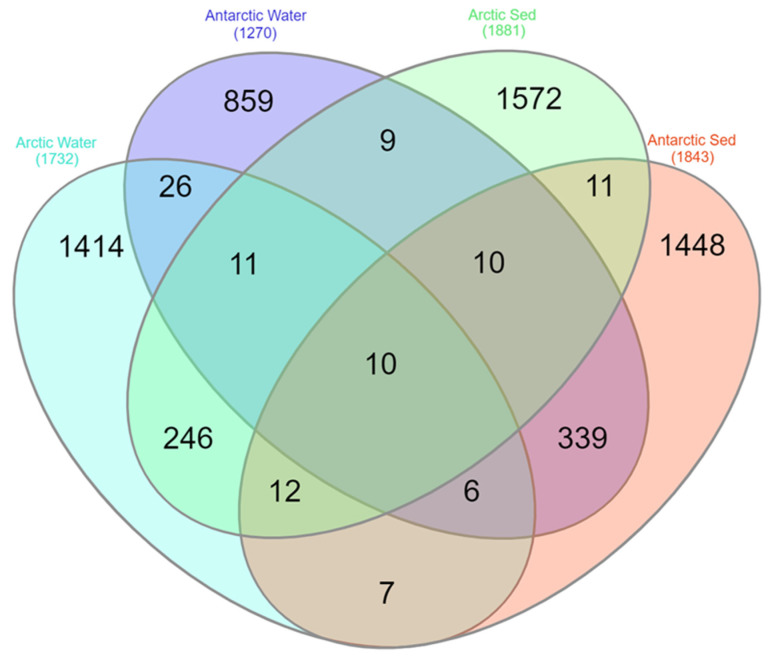
Venn diagram generated using all the retrieved ASVs separated by region (Arctic and Antarctic) and matrix of origin (water and sediment). The diagram was made by InteractiVenn online tool [35].

**Table 1 jof-09-01095-t001:** Geographical and physical–chemical data for each sampling site.

Area	Lake	Sample ID	Coordinates		Physical-Chemical Parameters			
					Water Temperature(°C)	pH	O_2_ %	Cond (uS/cm)
Ny-Ålesund(Arctic)	Solvannet	L1	78°55′33.121″ N	11°56′19.618″ E	6.53	8.19	100.77	398
	Glacier	L2	78°55′2.64″ N	11°47′26.52″ E	8.83	7.85	98.67	150
	Knudsenheia	L3	78°56′40.801″ N	11°51′34.74″ E	9.00	8.40	106.50	2660
	Storvatnet	L4	78°55′27.181″ N	11°52′43.68″ E	7.90	8.09	102.67	241
	Tvillingvatnet	L5	78°55′3.4788″ N	11°51′55321″ E	8.13	7.82	102.47	222
Livingston Island(Antarctica)	Argentina	LA	62°40′22.39″ S	60°24′18.12″ W	0.20	5.56	66.00	65
	Sofia	LS	62°40′12.19″ S	60°23′17.90″ W	0.33	5.47	84.48	20
Deception Island(Antarctica)	Balleneros	LB	62°58′51.1″ S	60°34′27.1″ W	6.04	3.60	83.80	423
	Telefon	LT	62°55′39.9″ S	60°41′21.3″ W	7.37	6.03	86.12	467
	Zapatilla	LZ	62°59′00.24″ S	60°40′29.07″ W	6.80	5.30	84.65	81

**Table 2 jof-09-01095-t002:** Genera retrieved at a percentage above 1% in Arctic and Antarctic regions.

		Arctic								Antarctic									
Phylum	Genus	Solvannet	Glacier	Knudsenheia		Storvatnet		Tvillingvatnet		Argentina		Sofia		Balleneros		Telefon		Zapatilla	
		L1-w	L2-w	L3-w	L3-s	L4-w	L4-s	L5-w	L5-s	LA-w	LA-s	LS-w	LS-s	LB-w	LB-s	LT-w	LT-s	LZ-w	LZ-s
*Ascomycota*	*Alatospora*																		
	*Alfaria*																		
	*Alternaria*																		
	*Antarctolichenia*																		
	*Arthroderma*																		
	*Aspergillus*																		
	*Beauveria*																		
	*Cadophora*																		
	*Candida*																		
	*Cladosporium*																		
	*Coleophoma*																		
	*Collophora*																		
	*Coniosporium*																		
	*Debaryomyces*																		
	*Geomyces*																		
	*Haptocillium*																		
	*Helicodendron*																		
	*Heydenia*																		
	*Hyaloscypha*																		
	*Iodophanus*																		
	*Knufia*																		
	*Mastodia*																		
	*Metschnikowia*																		
	*Nectriopsis*																		
	*Neobulgaria*																		
	*Penicillium*																		
	*Polyphilus*																		
	*Pseudeurotium*																		
	*Pseudogymnoascus*																		
	*Scolecolachnum*																		
	*Tetracladium*																		
	*Thelebolus*																		
	*Ulvella*																		
*Basidiomycota*	*Amylocorticiellum*																		
	*Camptobasidium*																		
	*Coriolopsis*																		
	*Cryolevonia*																		
	*Cryptococcus*																		
	*Cutaneotrichosporon*																		
	*Cystofilobasidium*																		
	*Glaciozyma*																		
	*Hyphodermella*																		
	*Leucosporidium*																		
	*Malassezia*																		
	*Mrakia*																		
	*Naganishia*																		
	*Phenoliferia*																		
	*Pseudobensingtonia*																		
	*Scopuloides*																		
	*Sidera*																		
	*Vishniacozyma*																		
*Chytridiomycota*	*Betamyces*																		
	*Entophlyctis*																		
	*Lobulomyces*																		
	*Zygophlyctis*																		
*Mortierellomycota*	*Entomortierella*																		
	*Mortierella*																		

Abundance range in percentage value (%). 
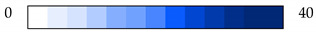

**Table 3 jof-09-01095-t003:** Diversity indices calculated using the total retrieved ASVs.

Polar Region	Matrice	Sample	Observed	Chao1	ACE	Shannon (H′)	Simpson	InvSimpson	Fisher
Arctic	Water	L1-w	274	274.38	275.00	2.89	0.88	8.38	37.17
		L2-w	563	563.40	564.21	4.80	0.98	53.50	80.69
		L3-w	184	185.75	186.92	1.72	0.57	2.34	22.62
		L4-w	585	587.02	589.17	4.10	0.95	19.78	86.61
		L5-w	470	471.49	473.39	2.99	0.86	7.40	66.10
		mean	415.2	416.41	417.74	3.3	0.85	18.28	58.64
	Sediment	L3-s	461	461.63	461.93	3.17	0.83	5.80	64.12
		L4-s	645	645.32	645.81	4.41	0.96	26.24	95.50
		L5-s	885	885.68	886.39	4.69	0.97	29.46	132.98
		mean	663.7	664.21	664.71	4.09	0.92	20.5	97.54
Antarctica	Water	LA-w	102	111.55	113.20	2.45	0.85	6.58	10.86
		LB-w	489	489.08	489.32	4.01	0.95	19.44	66.15
		LS-w	390	390.75	390.92	4.82	0.98	43.99	54.75
		LT-w	222	222.00	222.00	3.34	0.89	9.44	26.57
		LZ-w	243	244.91	244.85	2.55	0.83	5.83	30.79
		mean	289.2	291.66	292.06	3.43	0.9	17.06	37.83
	Sediment	LA-s	636	636.53	637.06	3.98	0.92	13.33	89.92
		LB-s	409	409.06	409.39	3.87	0.95	20.38	55.38
		LS-s	481	481.33	481.97	4.49	0.97	30.32	66.56
		LT-s	186	186.00	186.00	2.79	0.85	6.68	21.76
		LZ-s	398	398.13	398.37	3.80	0.94	15.41	53.39
		mean	422	422.21	422.56	3.79	0.93	17.22	57.4

## Data Availability

All sequences have been submitted to the National Center for Biotechnology Information (NCBI) and are associated to the BioProject PRJNA1000778.

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
