# Peer review of "A Deep Insight into the Diversity of Microfungal Communities in Arctic and Antarctic Lakes"

_jof, 2023, doi:10.3390/jof9111095_

Round 1

Reviewer 1 Report

Comments and Suggestions for Authors

Dear Authors,

Comments on this paper are as follows;

1. section2-1. Was the sample stored in RNA later or in a sample protector?

2. section2-2. Was the DNA extracted after it was brought back to the lab?

Or did you perform the DNA extraction at the resarch station?

Please add to the text.

3. section2-2. Sequencing by Miseq, did you do paired-end sequencing?

Also, did you use V3 reagents?

4. section3-2. Please add in the text what is the average length of DNA after assembly.

The following comments are personal opinions and are not a request to revise the manuscript.

 Even in the high latitude Arctic archipelago of Svalbard, there are effects of global warming.

A discussion of the effects of global warming and the diversity of fungi would make for a very interesting paper. 

Author Response

Dear Reviewer, 

thank you for your comments.

You can find our reply i attached file.

Best Regards

Reviewer 2 Report

Comments and Suggestions for Authors

The work presented by Marchetta et al. studies the microfungal diversity in water and sediment samples from both Arctic and Antarctic lakes, which is very interesting. To achieve this goal, a metabarcoding approach coupled with several statistical and diversity analyses were performed, revealing remarkable differences between both polar regions along with common features. I recommend accepting the manuscript after the following revisions.

Specific comments

Lines 57-62: This sentence is too long, please split it up.

Table1: Please, provide more detail in the materials and methods section on how this physical-chemical data were collected.

Lines 141-142: What filters were applied? Please, clarify this point.

Line 173: “conductivity”

Lines 184-185: The meaning of this sentence may be so that it is better to replace "also if" by "even though".

Line 187: What is the meaning of Fungi_phy_Incertae_sedis? Please, clarify it.

Line 224: "statistical analysis of the variance (ANOVA)". Please, provide more details in the materials and methods section on how this analysis was accomplished. Also, where are the results derived from these analyses?

Line 236: L4 and L3 seem to be more related to Rozellomycota. Please, clarify it.

Line 247: Please, check "Amylocorticiellum and Leucosporidium were only found in L5-s" as it seems as if the latter was present in more lakes.

Line 255: "exception for LT and LB-s"

Line 239: What cutoff was used to set the genus level?

Line 258: "even though"

Line 291: Please, provide the information regarding the Antarctic lake into a new paragraph.

Lines 303-306: Where is this result represented? Please, clarify it.

Table 3, Line 329: It would add more value to include the mean of these indices for Water and Sediment in each polar region.

Line 364: Please, provide this H' value in Table 3.

Lines 399-456: Please, consider the addition of more paragraphs as it will ease the reading.

Line 483: The line starting with "In our study..." should go to a different paragraph.

Comments on the Quality of English Language

NA

Author Response

Dear Reviewer, 

thank you for your comments.

You can find our reply i the attached file.

Best Regards
